# A THEORETICALLY GROUNDED CHARACTERIZATION OF FEATURE REPRESENTATIONS

## ABSTRACT

A large body of work has explored how learned feature representations can be useful for a variety of downstream tasks. This is true even when the downstream tasks differ greatly from the actual objective used to (pre)train the feature representation. This observation underlies the success of, e.g., few-shot learning, transfer learning and self-supervised learning, among others. However, very little is understood about why such transfer is successful, and more importantly, how one should choose the pre-training task. As a first step towards this understanding, we ask: what makes a feature representation good for a target task? We present simple, intuitive measurements of the feature space that are good predictors of downstream task performance. We present theoretical results showing how these measurements can be used to bound the error of the downstream classifiers, and show empirically that these bounds correlate well with actual downstream performance. Finally, we show that our bounds are practically useful for choosing the right pre-trained representation for a target task.

## 1 INTRODUCTION

Since the (re)-discovery of neural networks for visual recognition, a plethora of work has observed that the feature representations trained on ImageNet generalize to many downstream tasks, even to new domains (Donahue et al., 2014; Kornblith et al., 2019; Kolesnikov et al., 2020; Wallace & Hariharan, 2020). This observation, and the resulting gain in accuracy even with very limited labels, has heralded new research directions into *other* ways of learning representations, such as *self-supervised learning* (Chen et al., 2020) or meta-learning (Snell et al., 2017).

This growing field of *representation learning* has yielded ostensibly better and better feature representations. However, a closer look reveals many mysterious results. For example, meta-learning methods do not transfer across domains (Guo et al., 2020) and self-supervised representations struggle with fine-grained recognition (Wallace & Hariharan, 2020). These empirical results are extremely valuable, but do not provide a deeper understanding of the corresponding phenomena. On the other side of the spectrum, theoretical work on neural representations is illuminating, but often makes assumptions about models or tasks (Du et al., 2020; Arora et al., 2019b). We lack a general understanding of which neural representations work well for a given task and why.

In this paper, we take a first step towards such an understanding by developing lower and upper bounds on classifier accuracy based on *data-driven* properties of the *feature space*. Classical theoretical bounds focus entirely on the complexity of the classifier and ignore the feature space. In contrast, our bounds are based on two intuitive properties of the feature representation (Figure 1): (a) **Local alignment**, which is the degree to which nearby data points share labels, and (b) **local congregation** which is the degree to which data points embed close to each other. Intuitively, if the feature representation is locally aligned to the task, then any smooth classifier will be able to model the task well. If it is additionally congregated, then most test points will have nearby training instances, so the classifier will generalize well from limited data. We show that our bounds are not only intuitive and theoretically justified, but also *predictive* of actual performance in practical settings: on a large dataset of realistic few-shot tasks, we can use our bounds to pick the best pre-trained representation without any training. Taken together, our work is a first step towards a general, intuitive characterization of the feature space that is *predictive* of downstream classifier performance.

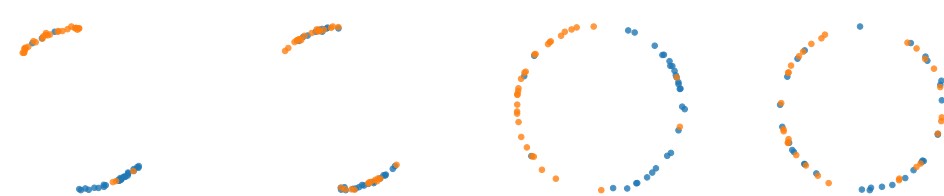

Figure 1: An illustration of the properties of local alignment and congregation. Feature representations on the left are more congregated than those on the right. The first and third feature representations on the top are more aligned locally than the other two. Classical error bounds depend only on the norm of the feature vectors and so cannot distinguish between these. We show theoretically and empirically that the distinctions shown here impact the error of a downstream classifier.

## 2 RELATED WORK

**Analyzing transferability:** The motivations of our work lie in understanding the empirical effectiveness of transferring feature representations from supervised (Donahue et al., 2014; Zhai et al., 2019; Kolesnikov et al., 2020) or self-supervised (Goyal et al.; Wallace & Hariharan, 2020; Chen et al., 2020) tasks. With these successes on transfer, and with the availability of a large number of pretrained features trained on a wide variety of domains, there has been an increasing interest in *predicting transferability*, or in *selecting the right features* for a particular target task. The simplest approach is to train a classifier with every available pre-trained representation and pick the best performer (Zamir et al., 2018; You et al., 2021). This is both computationally expensive and requires lots of labeled data for the target task. If the pre-training task is a classification task with a known label space then the conditional distribution of targets given pre-training task labels is informative of transfer performance (Tran et al., 2019; Nguyen et al., 2020). However, this approach is difficult to apply if the pre-training task is not classification, or is inaccessible (as with models trained on proprietary data). This inaccessibility of pre-training data is also a problem for approaches that match pre-training and target distributions (Gao & Chaudhari, 2021), or explicitly adapt the pre-training images or label space (Cui et al., 2018). In contrast to these approaches, we focus on directly analyzing the pre-trained feature representation, which allows us to be agnostic to the actual task it was trained on. Our work is most similar in setup and motivation to work on measuring the alignment between feature representations and the target labels (Huang et al., 2021; Bao et al., 2019), but in addition to allowing model selection, yields intuitive and general bounds on accuracy. Our work is orthogonal to work on characterizing tasks and measuring task similarity; this frequently requires pre-trained features in the first place (Achille et al., 2019; Wallace et al., 2021; Song et al., 2020; Dwivedi et al., 2020; Dwivedi & Roig, 2019).

**Analyzing feature representations:** Our work is also related to research on understanding feature representations in general. Some of this work has focused on understanding invariance properties of convnet features (Aubry & Russell, 2015). Others have looked at what individual feature dimensions mean (Agrawal et al., 2014; Zeiler & Fergus, 2014; Szegedy et al., 2013). Still others have explored if and when features from pre-trained networks transfer well between tasks as a function of the layer chosen (Yosinski et al., 2014) or the architecture (Kornblith et al., 2019). Recently these explorations have been extended to other representation learning techniques, notably self supervised techniques (Wang & Isola, 2020; Wallace & Hariharan, 2020). The insights from these explorations inspire our results. However, these explorations have been primarily empirical, and are therefore limited by the diversity of real world benchmarks that they experiment with. On the other end of the spectrum, there is prior work on theoretical analysis of transfer learning. This prior work often follows the framework proposed by Baxter (2000), and in so doing makes assumptions about the distributions of the different tasks (Maurer, 2009; Du et al., 2020; Galanti et al., 2016; Pentina & Lampert, 2014). In contrast, our approach eschews these assumptions in lieu of data-driven measurements. Data-driven complexity measures have been explored before for analyzing neural network training dynamics and generalization. These measures include eigenvectors of the gram matrix between data points (Arora et al., 2019a), the variance and separation of class-specific manifolds (Cohen et al., 2020), the underlying intrinsic dimensionality of the task (Lampinen & Ganguli, 2018) or the mutual information between representations and the inputs or labels (Shwartz-Ziv &

Tishby, 2017). The corresponding results can be used for analyzing feature representations as well. Our proposed measurements are similar, but are simpler to measure and potentially more intuitive.

## 3 PROBLEM SETUP

Suppose we are interested in mapping a space of inputs $\mathcal{X}$ to a space of targets $\mathcal{Y}$. There is an underlying distribution $\mathcal{D}$ over $\mathcal{X} \times \mathcal{Y}$. We have a feature representation $\phi : \mathcal{X} \to \mathbb{R}^f$ which might be pre-trained on another dataset or task, which is inaccessible to us. We will assume that $\|\phi(x)\| \leq B$.

For ease of exposition in the paper, we focus on the task of binary classification; the case of multi-class classification is similar. For binary classification, we write the label space as $\mathcal{Y} = \{-1, 1\}$. Our classifier will use a scoring function that operates on feature space, $h : \mathbb{R}^f \to \mathbb{R}$. The predicted label will then be $\text{sign}(h(\phi(x)))$, where if $h(\phi(x))$ is 0, we will arbitrarily assign a label of -1. The set of possible functions $h$ defines the hypothesis class for the classifier; denote this by $\mathcal{H}$. For most of our analysis, we primarily care about the *smoothness* of the functions in $\mathcal{H}$. We will assume that all functions in $\mathcal{H}$ are Lipschitz continuous with Lipschitz constant less than $W$. Thus:

$$h(\phi(x)) - h(\phi(x')) \leq W\|\phi(x) - \phi(x')\| \quad \forall h \in \mathcal{H} \tag{1}$$

We note that one such hypothesis class which is commonly used in practice is the class of *linear classifiers of bounded norm*: $\mathcal{W} = \{\mathbf{v} \mapsto \mathbf{w}^T\mathbf{v}; \|\mathbf{w}\| \leq W\}$.

Because the zero-one loss, $l^*(h(\phi(x)), y) = \mathbb{I}[\text{sign}(h(\phi(x))) \neq y]$ is difficult to analyze, we will use the following continuous upper bound which is standard in theoretical treatments (Mohri et al., 2018)

$$l(h(\phi(x)), y) = \min(1, \max(0, 1 - yh(\phi(x)))) \geq l^*(h(\phi(x)), y) \tag{2}$$

Our focus in this paper is to understand how the properties of the feature extractor $\phi$ affect the loss of the classifiers $l(h(\phi(x)))$. In particular, we wish to understand how $\phi$ impacts (a) the *lowest* average error that one can achieve, and (b) the *true error* when one generalizes from a small training set.

We begin by first identifying the key properties of feature representations.

## 4 TWO PROPERTIES OF FEATURE REPRESENTATIONS

What properties should we use to characterize feature representations? First, we should use properties that are easy to measure, potentially with limited labeled data. Second, these properties should be easy for human developers and practitioners to reason about. Third, they should correlate well with the final accuracy of downstream classifiers. In sum, we want simple, intuitive measurements of the feature space that are predictive of downstream accuracy.

One kind of intuitive measurement is to look at what the feature representation considers as similar. In particular, we could look at pairs of examples that embed close to each other in feature space and ask if they are indeed similar in terms of their ground truth labels. We call this property **local alignment**. In particular, we make the following definition

**Definition 1.** *Suppose $\alpha > 0$. The **local alignment** of the feature space $\phi$, denoted by $p_a^\phi(\alpha)$ is the probability that two data points $(x, y), (x', y') \sim \mathcal{D}$ share a label given that they embed within a distance of $\alpha$:*

$$p_a^\phi(\alpha) \triangleq P(y = y' \mid \|\phi(x) - \phi(x')\| \leq \alpha, (x, y), (x', y') \sim \mathcal{D}) \tag{3}$$

$\alpha$ here is a hyperparameter which governs the resolution at which we do the analysis. Note that this notion of local alignment is different from the alignment proposed by Wang & Isola (2020): the latter looks at how often two data points that are semantically similar embed close to each other, while the former looks at how often two data points that embed close to each other are semantically similar. A feature space that is locally aligned per our definition may not actually be aligned per the definition of Wang & Isola (2020) because two very similar images may be embedded far away from each other.

Local alignment alone may be meaningless if data points do not generally embed close to each other. We also need the feature space to be such that data points generally **congregate**:

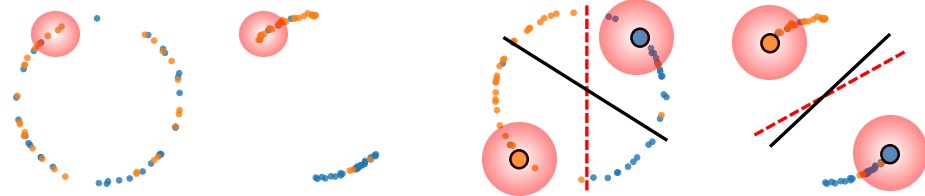

Figure 2: Intuition behind our analysis for the lower bound on error (Theorem 1, Left) and upper bound on generalization (Theorem 2, Right). In each case we look at pairs of data points that fall within a distance $\alpha$ of each other (shown as red balls). The lower bound notes that examples with different labels in these balls yield high error. The upper bound notes that outside the red balls centered on training points (large dots) generalization is not guaranteed. Also shown are the optimal (red) and obtained (black) classifiers.

**Definition 2.** *Suppose $\alpha > 0$. The **degree of congregation** of the feature space $\phi$, denoted by $p_c^\phi$, is the probability that two points $x, x'$ sampled from $\mathcal{D}$ embed within a distance of $\alpha$:*

$$p_c^\phi(\alpha) \triangleq P(\|\phi(x) - \phi(x')\| \le \alpha | x, x' \sim \mathcal{D}) \tag{4}$$

We have $\alpha$ as a hyperparameter here too.

We now use these notions of *local alignment* and *congregation* to analyze the downstream error of classifiers.

## 5 WHAT IS THE BEST ACCURACY WE CAN ACHIEVE?

Given a feature space $\phi$, what is the best accuracy we can hope to get using scoring functions from $\mathcal{H}$? Recall that these scoring functions are all $W$-Lipschitz. Our key intuition is that this Lipschitz continuity will force any scoring function in $\mathcal{H}$ to produce very similar scores for nearby data points. If these have different labels, then the classifier will be forced to err on at least one of them. To formalize this intuition, we begin with the following claim

**Claim 1.** *Consider two data points $(x, y)$ and $(x', y')$. If $\|\phi(x) - \phi(x')\| < \frac{1}{W}$ and $y \ne y'$, then:*

$$l(h(\phi(x)), y) + l(h(\phi(x')), y') \ge 1 \tag{5}$$

*Proof.* Observe first that due to Lipschitz continuity of $h$:

$$|h(\phi(x)) - h(\phi(x'))| \le W \|\phi(x) - \phi(x')\| \le 1 \tag{6}$$

$$\Rightarrow h(\phi(x')) > h(\phi(x)) - 1 \tag{7}$$

Now, since $y \ne y'$, it follows that $y' = -y$. Without loss of generality, let us assume that $y = 1$. Denote $l(h(\phi(x)), y)$ by $a$ and $l(h(\phi(x')), y')$ by $b$. We will prove this claim by contradiction. Suppose, if possible, that the clam is not true, and $a + b < 1$. Since the loss is always non-negative, it follows that $a < 1$ and $b < 1$. Thus:

$$a = \max(0, 1 - yh(\phi(x))) \tag{8}$$

$$\ge 1 - yh(\phi(x)) = 1 - h(\phi(x) \tag{9}$$

$$\Rightarrow h(\phi(x)) \ge 1 - a \tag{10}$$

Taking the other data point:

$$b = \max(0, 1 - y'h(\phi(x'))) \tag{11}$$

$$\ge 1 - y'h(\phi(x')) = 1 + h(\phi(x')) \tag{12}$$

$$\ge 1 + h(\phi(x)) - 1 = h(\phi(x)) \qquad (\because equation\ 7) \tag{13}$$

$$\ge 1 - a \tag{14}$$

$$\Rightarrow a + b \ge 1 \tag{15}$$

thus yielding a contradiction. $\qquad\qquad\square$

Thus, if two data points are close, at least one of them must be incorrectly classified. We can thus bound the error of the best possible classifier from below by estimating how often this happens:

**Theorem 1.** *Let $l$ be the loss function defined above, and $\mathcal{H}$ be a hypothesis class of Lipschitz functions with Lipschitz constant at most $W$. Let $\mathcal{D}$ be a distribution over $\mathcal{X} \times \mathcal{Y}$, and $p_a^\phi$ and $p_c^\phi$ defined as above. Then*

$$\inf_{h \in \mathcal{H}} \mathbb{E}_{x,y \sim \mathcal{D}}[l(\mathbf{h}(\phi(x)), y)] > \frac{1}{2} \left( 1 - p_a^\phi \left( \frac{1}{W} \right) \right) p_c^\phi \left( \frac{1}{W} \right) \tag{16}$$

*Proof.* We observe that if $(x, y), (x', y') \sim \mathcal{D}$, then the probability that $\|\phi(x) - \phi(x')\| < \frac{1}{W}$ and $y \neq y'$ is given by $\tilde{p} = (1 - p_a^\phi(1/W))p_c^\phi(1/W)$. Then, for all $h \in \mathcal{H}$.

$$\mathbb{E}_{x,y \sim \mathcal{D}}[l(h(\phi(x)), y)] = \frac{1}{2} \mathbb{E}_{(x,y),(x',y') \sim \mathcal{D}}[l(h(\phi(x)), y) + l(h(\phi(x')), y')] \tag{17}$$

$$\geq \frac{\tilde{p}}{2} \mathbb{E}[l(h(\phi(x)), y) + l(h(\phi(x')), y') \mid \|\phi(x) - \phi(x')\| < \frac{1}{W} \text{ and } y \neq y'] \tag{18}$$

$$\geq \frac{\tilde{p}}{2} \tag{19}$$

$$\tag{20}$$

where the second step follows because the loss is non-negative, so only focusing on the case when $x$ and $x'$ are close but have different labels yields a lower bound. This is true for all $h \in \mathcal{H}$, so it is true for the infimum as well. $\square$

# 6 How well will classifiers generalize?

There may be classifiers that yield low error, but it can be challenging to find these from small training sets. Therefore we now ask if we can use the feature space to analyze how well classifiers trained on small training sets might *generalize*.

While generalization bounds abound, they typically do not include any reasoning about the underlying feature space beyond the maximum norm of feature vectors. As such, these bounds are insufficient for understanding the impact of the feature space on generalization.

Here, we present two kinds of bounds. The first bound uses a similar proof strategy of relying on the Lipschitz continuity of the classifier. The second bound leverages a more traditional Rademacher complexity-based analysis to bound the excess risk (i.e., the difference between the test loss and training loss).

## 6.1 Bounding the probability of high-loss data points

Classical generalization bounds are based primarily on concentration inequalities and the law of large numbers. This produces substantially coarse estimates for applications like *few-shot learning*, where the training dataset has very few labels.

We propose a simpler alternative bound here. We observe that owing to the Lipschitz continuity of the classifier, test data points that are close to training data points will have very similar loss as the corresponding training points, which can be no worse than the loss on the worst performing training example.

**Theorem 2.** *Let $\mathcal{H}_\phi = \{z = x \mapsto h(\phi(x)); h \in \mathcal{H}\}$. Suppose $S$ is a sampled training set of $m$ points. For any $z \in \mathcal{H}_\phi$, let $l_{max}(z, S) = \max_{(x,y) \in S} l(z(x), y)$ be the maximum loss $z$ incurs on $S$. Then, for all $\epsilon > 0$ and $(x, y) \sim \mathcal{D}$:*

$$P\left( l(z(x), y) > l_{max}(z, S) + \epsilon \right) \leq \left( 1 - p_a^\phi \left( \frac{\epsilon}{W} \right) p_c^\phi \left( \frac{\epsilon}{W} \right) \right)^m . \tag{21}$$

*Proof.* It can be shown that for any $y$, $l(z(x), y) - l(z(x'), y) \leq |z(x) - z(x')|$ (see Appendix). Further, $\forall h \in \mathcal{H}$, $h$ is $W$-Lipschitz. So we have, for all $x, x', y$:

$$l(z(x), y) - l(z(x'), y) \leq |z(x) - z(x')| \leq W\|\phi(x) - \phi(x')\| \tag{22}$$

It follows that for any $(x, y), (x', y')$:

$$\left. \begin{array}{l} \|\phi(x) - \phi(x')\| < \frac{\Delta}{W} \\ y = y' \end{array} \right\} \Rightarrow l(z(x), y) \leq l(z(x'), y') + \Delta \qquad \forall z \in \mathcal{H}_\phi \tag{23}$$

Note that the probability of sampling such an $(x', y')$ is $p_a^\phi(\Delta/W)p_c^\phi(\Delta/W)$. If $(x', y')$ is in $S$, then $l(z(x'), y') \leq l_{max}(z, S)$. Thus, for any $(x, y) \sim \mathcal{D}$

$$P\left(l(z(x), y) \leq l_{max}(z, S) + \epsilon\right) \geq P\left(\exists (x', y') \in S \text{ s.t } \|\phi(x) - \phi(x')\| < \frac{\Delta}{W} \text{ and } y = y'\right) \tag{24}$$

$$= 1 - \left(1 - p_c^\phi\left(\frac{\epsilon}{W}\right)p_a^\phi\left(\frac{\epsilon}{W}\right)\right)^m \tag{25}$$

$\square$

## 6.2 BOUNDING THE EXCESS RISK

We can also perform a more traditional analysis of the excess risk, which uses the law of large numbers and various concentration inequalities. It is well known that the difference between train loss and the test loss is upper bounded by the Rademacher complexity of the hypothesis class plus a small constant (Mohri et al., 2018). Concretely, with probability greater than $1 - \delta$, the following holds for all $h \in \mathcal{H}$:

$$R(h) - \hat{R}(h, S) \leq \mathcal{R}_m(\mathcal{H}) + \sqrt{\frac{\log \frac{1}{\delta}}{2m}} \tag{26}$$

where $R(h)$ is the test loss (or expected risk), $\hat{R}(h, S)$ is the training loss (or empirical risk), and $\mathcal{R}_m(\mathcal{H})$ is the *Rademacher complexity* of $\mathcal{H}$.

Thus, to bound the generalization error, it suffices to bound the Rademacher complexity of $\mathcal{H}$, which is defined as $\mathcal{R}_m(\mathcal{H}) = \mathbb{E}_{S \sim \mathcal{D}^m, \sigma} \sup_{h \in \mathcal{H}} \frac{1}{m} \sum_{i=1}^m \sigma_i h(x_i)$. Here $\sigma_i$ are Rademacher random variables (i.e., $\sigma_i \in \{-1, 1\}, \mathbb{E}\sigma_i = 0$). The Rademacher complexity is a measure of the size of the hypothesis class, and can be used to bound the difference between a hypothesis' performance on two sampled datasets. To allow us to define a concrete bound, we will analyze the class of *linear classifiers with bounded norm*, $\mathcal{W}$.

The key insight here is that if, for a particular $\alpha$, $p_c^\phi(\alpha)$ is large, then any two datasets (e.g., a train and a test dataset) that we sample from the underlying distribution will be "similar", in terms of having nearby data points. As such, linear classifiers will be forced to produce similar scores for both datasets, yielding similar loss. This insight is expressed as the following bound on the Rademacher complexity of the set of linear classifiers composed with $\phi$:

---

**Theorem 3.** *Let $\mathcal{W}_\phi = \{z = x \mapsto h(\phi(x)); h \in \mathcal{W}\}$. Let $p_c^\phi$ be defined as above. Then the Rademacher complexity of $\mathcal{W}_\phi$ is bounded above by:*

$$\mathcal{R}_m(\mathcal{W}_\phi) \leq \frac{W\left(\alpha\sqrt{p_c^\phi(\alpha)/2} + 2B\sqrt{(1 - p_c^\phi(\alpha)) + 2(1 - p_c^\phi(\alpha))^2}\right)}{2\sqrt{m}} \tag{27}$$

---

*Proof sketch*: (Full proof presented in Appendix). For any $x_i \in S$, there is a probability $p_c^\phi(\alpha)/2$ that the next point $x_{i+1}$ is (a) within a distance $\alpha$ in feature space, and (b) is multiplied with a Rademacher variable of the opposite sign. Such pairs will effectively cancel out each other except for a small quantity of $\alpha$, resulting in the first term. The second term is obtained from all the rest of the data points in $S$, which have to be analyzed in the usual way.

## 7 IMPLICATIONS

Both the lower and upper bounds (in particular theorem 2) suggest the need for a high $p_a$, namely, a feature space where nearby examples have similar labels. However, the two bounds *differ* in the need for congregation.

The lower bound suggests that we need a *less congregated* feature space, i.e., a space where examples are in general far apart. This matches the training objective of self-supervised and contrastive learning techniques, which primarily attempt to push examples apart. This is demonstrated in Figure 3, which shows that ImageNet training and self-supervised training produce not just *locally aligned* (i.e., high $p_a$), but also *non-congregated* feature representations (i.e, low $p_c$).

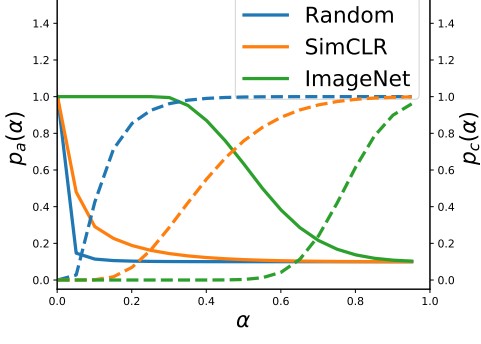

Figure 3: $p_a$ (left axis, solid line) and $p_c$ (right axis, dotted line) as a function of $\alpha$ for three feature representations on CIFAR-10

However, the upper bound suggests that good generalization from small datasets actually requires a *highly congregated feature space*. As such, these highly spread out feature representations can actually yield large generalization errors in few-shot settings, as demonstrated in Table 1, even if they yield much lower errors with large training sets. Thus, finding a good representation for a task is a nuanced decision, requiring one to balance between these conflicting requirements.

**Theorem 2 vs 3**: The two generalization bounds both demand high congregation, but differ in whether they involve alignment. Interestingly, the Rademacher complexity-based bound does not use $p_a$. Below, we will see that for this reason, this bound is less correlated with the downstream performance.

## 8 EMPIRICAL ANALYSIS

Above we have shown that $p_a^\phi$ and $p_c^\phi$ yield bounds on classifier performance. But are they useful enough in practice to make decisions? We first look to see whether they are correlated with actual loss values, and then present potential applications.

| Representation | $p_a$ | $p_c$ | Test loss (large dataset) | Excess risk (5-shot) |
|---|---|---|---|---|
| Random | 0.51 | 0.88 | 0.29 | 0.26 |
| SimCLR | 0.7 | 0.1 | 0.18 | 0.24 |
| ImageNet | 1.0 | 0.0 | 0.04 | 0.65 |

Table 1: Three representations with different alignment and congregation, and the corresponding test loss (with large training sets) as well as excess risk with tiny training sets.

### 8.1 CORRELATIONS WITH CLASSIFIER ACCURACY

We used CIFAR-10(Krizhevsky, 2009) as our test bed. We sampled pairs of classes from CIFAR-10 as "recognition tasks", resulting in 45 different tasks. For each task we ran 10 different trials. In each trial, we sampled a small 20-example training set to train a linear classifier, and then evaluated the classifier on a large validation set. The training and validation margin losses were averaged over the 10 trials. We used 18 different representations, all normalized to have maximum norm 1, and all with the architecture of a ResNet-18. These representations were trained with labels of varying granularity on ImageNet(Russakovsky et al., 2015), iNaturalist(Van Horn et al., 2018), or CIFAR-100 (see Appendix). For each task and for every representation, we computed our bounds using the labeled validation set to get $p_a^\phi$ and $p_c^\phi$, and using the empirically obtained classifier norm as $W$ (a more practical approach is described in the next section).

**Results:** For each task, we computed the correlations between our proposed bounds and the corresponding true losses. Histograms of these correlations across the 45 tasks are shown in Figure 4 (scatter plots with individual data points are in Figure 6). We find that the lower bound is extremely

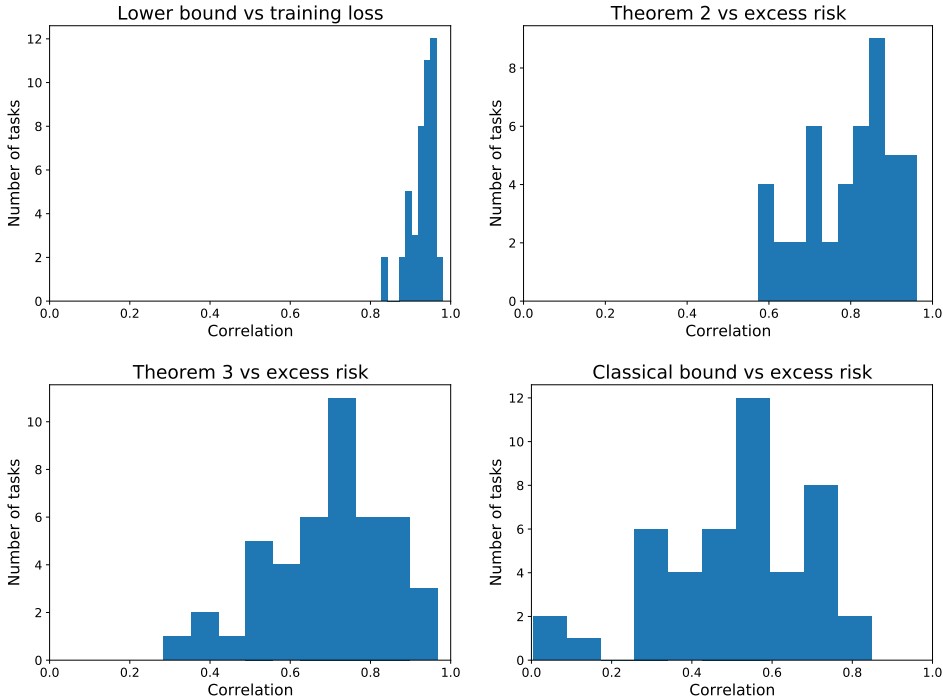

Figure 4: Correlations of our proposed bounds with actual margin loss. Clockwise from top left: Lower bound vs train loss, Theorem 2 vs excess risk, the classical textbook bound vs excess risk, and Theorem 3 (right) vs actual excess risk.

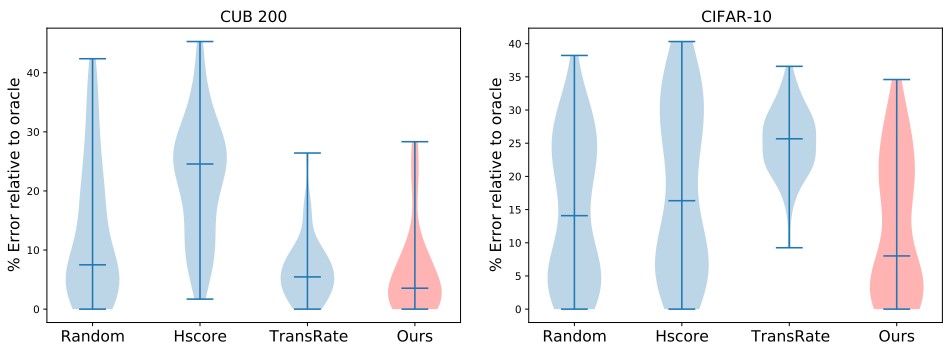

Figure 5: Our characterization of the feature space allows us to choose the best pre-trained representation for a given few-shot task. Lower numbers are better. Left: results on CUB-200. Right: results on CIFAR-10.

well correlated with the training loss (average Spearman's $\rho$ 0.93), but less so with the test loss (average Spearman"s $\rho$ 0.55). This is of course expected because the lower bound does not take into account the perils of classifiers trained on small training sets. The difference between the training and test loss correlates very well with our proposed upper bound in Theorem 2 (average Spearman's $\rho$ 0.80), which is much better than the correlation obtained with the classical bound (average $\rho$ 0.50). Theorem 3 yields a worse correlation that Theorem 2, perhaps because it doesn't take into account the alignment. Since Theorem 2 yields a stronger correlation, we use this upper bound in conjunction with the lower bound for a more practical experiment on model selection below.

## 8.2 APPLICATIONS TO FEW-SHOT TRANSFER

We now see if our bounds can help choose the right feature representation for problems with very limited labeled data. In particular, we assume that we have a very small labeled set $S$ (e.g., 10 labeled examples per class) and a larger unlabeled set $U$. Because labeled data is very limited, one cannot afford to use labeled data for held-out validation sets. Therefore, a way of characterizing feature representations without training and evaluating classifiers is valuable.

**Baselines:** Much of prior work on estimating transferability assumes that the representations come from classification tasks Nguyen et al. (2020) on accessible datasets Cui et al. (2018), or that an initial appropriate feature representation is available Achille et al. (2019); Dwivedi et al. (2020). With the advent of self-supervised learning Chen et al. (2020) and large proprietary datasets Kolesnikov et al. (2020), and the wide variety of visual domains Wallace & Hariharan (2020), these assumptions are no longer appropriate. We therefore consider only baselines that make minimal assumptions about the pre-trained feature representation and the target task: H-score Bao et al. (2019) and TransRate Huang et al. (2021). Both approaches are based on information-theoretic arguments justifying the use of intra- and inter-class covariance matrices, which are global statistics of the feature representation, in contrast to our more local measurements. In addition, we also include a baseline that chooses a representation at random.

**Experimental setup:** We run few-shot experiments on two datasets: CIFAR-10 and CUB-200(Welinder et al., 2010). The pre-trained representations were obtained by training on iNaturalist, ImageNet and CIFAR-10 (only for CUB-200) with various loss functions. In each case, we sample 50 binary classification problems with 10 labeled examples each; all other examples from the classes involved are unlabeled. To estimate $p_a^\phi(\alpha)$ using just 10 labeled examples, we choose $\alpha$ to be large enough that for at least one class, every pair of labeled examples is at most $\alpha$ away in feature space. Our estimate of transferability involves linearly combining our lower bound on error (with a weight of 100 because of its smaller scale) and the upper bound on generalization from Theorem 2; lower values indicate better representations. For the baselines, the unconditional covariance matrix was estimated using the unlabeled data and other label-dependent measurements used the labeled data. We evaluate all approaches in terms of the average accuracy drop relative to the oracle representation (the one which post-hoc yields the highest test accuracy). Thus lower drops are better.

**Results:** We show results in figure 5. Interestingly, TransRate and H-score seem to have opposite trends on these two datasets, with H-score performing worse than random selection for CUB and TransRate performing worse than random on CIFAR-10 Our approach consistently yields better choices than both baselines on both benchmark datasets thus suggesting the usefulness of our characterization for practical problems.

## 9 LIMITATIONS, CONCLUSION AND FUTURE WORK

We have presented here a characterization of feature representations that is predictive of how well downstream classifiers will do. Our analysis is limited to classification, though the results may be adapted potentially to other classification-like tasks (e.g., object detection). We also assume that the downstream classifier is linear, which is in line with how feature representations are typically evaluated (Goyal et al.); however practical systems may use more powerful classifiers where our bounds may not apply. Finally, while our bounds are interpretable and predictive, it is not clear if they can be used to drive training objectives. This is an important direction for future work.

**Reproducibility:** All theoretical proofs and experimental details are included in either the main paper or the appendix. Code and pre-trained representations will be made public upon acceptance.

**Ethics statement:** This work aims to improve our understanding of feature representations, thus paving the way for more widespread use of pre-trained feature representations. We hope that this makes the power of sophisticated visual recognition techniques available to a much broader community who may not have the data or resources to train big networks from scratch. However, we note that our work focuses only on the *average accuracy* one can obtain from these representations in a downstream task; we do not delve into the biases that these representations might perpetuate Steed & Caliskan (2021), or privacy or consent issues in the original datasets used to train these

representations Birhane & Prabhu (2021). We suggest that when choosing a pre-trained representation, practitioners should consider not just our characterization of downstream accuracy, but also a broader characterization in terms of the many ethical implications of choosing a representation pre-trained on questionably collected or opaque datasets.

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

## A  APPENDIX

### A.1  PROOFS

**Theorem 4.** *Let $f(x) = \min(1, \max(0, x))$. Then $f(x) - f(x') \le |x - x'|$.*

*Proof.* We do a case-by-case analysis.

1. $x \ge 0, x' \le 1$. In this case, $f(x') \ge x'$ and $f(x) \le x$. So $f(x) - f(x') \le x - x' \le |x - x'|$.

2. $x < 0$. In this case $f(x) = 0$. Since $f(x') \ge 0$, it follows that $f(x) - f(x') < 0 < |x - x'|$.

3. $x' > 1$. In this case $f(x') = 1$. Since $f(x) \le 1$ it follows that $f(x) - f(x') < 0 < |x - x'|$.

□

**Claim 2.** *(Used in proof of Theorem 2) Let $l(z, y) = \min(1, \max(0, 1 - zy))$. Then $l(z, y) - l(z', y) \le |z - z'|$.*

*Proof.* $l(z, y) = f(1 - yz)$, where $f$ is defined as above. The claim follows directly from Theorem 4.
□

**Theorem 5.** *Let $\mathcal{W}_\phi = \{z = x \mapsto h(\phi(x)); h \in \mathcal{W}\}$. Let $p_c^\phi$ be defined as above. Then the Rademacher complexity of $\mathcal{W}_\phi$ is bounded above by:*

$$\mathcal{R}_m(\mathcal{W}_\phi) \le \frac{W\left(\alpha\sqrt{p_c^\phi(\alpha)/2} + 2B\sqrt{(1 - p_c^\phi(\alpha)) + 2(1 - p_c^\phi(\alpha))^2}\right)}{2\sqrt{m}} \tag{28}$$

*Proof.* First note that:

$$\mathcal{R}_m(\mathcal{W}_\phi) = \mathbb{E}_{S,\sigma} \sup_{z \in \mathcal{W}_\phi} \frac{1}{m} \sum_{i=1}^m \sigma_i z(x_i) \tag{29}$$

$$= \frac{1}{m} \mathbb{E}_{S,\sigma} \sup_{h \in \mathcal{W}} \sum_{i=1}^m \sigma_i \mathbf{w}_h^T \phi(x_i) \tag{30}$$

$$= \frac{1}{m} \mathbb{E}_{S,\sigma} \sup_{h \in \mathcal{W}} \mathbf{w}_h^T (\sum_{i=1}^m \sigma_i phi(x_i)) \tag{31}$$

$$\le \frac{W}{m} \mathbb{E}_{S,\sigma} \| \sum_{i=1}^m \sigma_i \phi(x_i) \| \qquad (\because \|\mathbf{w}_h\| \le W) \tag{32}$$

$$\tag{33}$$

Thus we need to bound $\mathbb{E}_{S,\sigma} \| \sum_{i=1}^m \sigma_i \phi(x_i) \|$

Consider any set $S$ of $m$ points sampled from $\mathcal{D}$. Let $\pi$ be the mapping $i \mapsto (i + 1) \mod m$, and define $\sigma_i' = \sigma_{\pi(i)}$ and $x_i' = x_\pi(i)$. Then, for any $z \in \mathcal{H}_\phi$ we have that $\sum_{i=1}^m \sigma_i \phi(x_i) = \sum_{i=1}^m \sigma_i' \phi(x_i') = \frac{1}{2} \sum_{i=1}^m (\sigma_i \phi(x_i) + \sigma_i' \phi(x_i'))$.

Note that $x_i \perp\!\!\!\perp x_i'$ and $\sigma_i \perp\!\!\!\perp \sigma_i'$. Define $a_i = \mathbb{I}[(\sigma_i = -\sigma_i')$ and $(\|\phi(x_i) - \phi(x_i')\| < \alpha)]$. Then $\mathbb{E}_{S,\sigma} a_i = \mathbb{E}_{S,\sigma} a_i^2 = \frac{p_c^\phi(\alpha)}{2}$, which we will denote as $p_\alpha$.

Therefore:

$$\mathbb{E}_{S,\sigma} \| \sum_{i=1}^m \sigma_i \phi(x_i) \| \tag{34}$$

$$= \frac{1}{2} \mathbb{E}_{S,\sigma} \| \sum_{i=1}^m (\sigma_i \phi(x_i) + \sigma_i' \phi(x_i')) \| \tag{35}$$

$$= \frac{1}{2} \mathbb{E}_{S,\sigma} \| \sum_{i=1}^m (a_i \sigma_i (\phi(x_i) - \phi(x_i')) + \sum_{i=1}^m \bar{a}_i (\sigma_i \phi(x_i) + \sigma_i' \phi(x_i'))) \| \tag{36}$$

$$\le \frac{1}{2} \mathbb{E}_{S,\sigma} \| \sum_{i=1}^m a_i \sigma_i (\phi(x_i) - \phi(x_i')) \|$$

$$+ \frac{1}{2} \mathbb{E}_{S,\sigma} \| \sum_{i=1}^m \bar{a}_i \sigma_i \phi(x_i) \|$$

$$+ \frac{1}{2} \mathbb{E}_{S,\sigma} \| \sum_{i=1}^m \bar{a}_i \sigma_i' \phi(x_i') \| \tag{37}$$

Observe that $a_i \perp\!\!\!\perp \sigma_i$, and $a_i \perp\!\!\!\perp \sigma_i'$. Using this, the second and third terms are similar. The second term yields:

$$\frac{1}{2}\mathbb{E}_{S,\sigma}\|\sum_{i=1}^{m}\bar{a}_i\sigma_i\phi(x_i)\| \tag{38}$$

$$=\frac{1}{2}\mathbb{E}_{S,\sigma}\sqrt{\sum_{i,j=1}^{m}\bar{a}_i\bar{a}_j\sigma_i\sigma_j\phi(x_i)^T\phi(x_j)} \tag{39}$$

$$\leq\frac{1}{2}\sqrt{\sum_{i,j=1}^{m}\mathbb{E}_{S,\sigma}\bar{a}_i\bar{a}_j\sigma_i\sigma_j\phi(x_i)^T\phi(x_j)} \tag{40}$$

We note that if $i \neq j$ and $i \neq \pi(j)$ and $j \neq \pi(i)$ then $\bar{a}_i\sigma_i \perp\!\!\!\perp \bar{a}_j\sigma_j$. Thus, looking at the expression under the square root:

$$\sum_{i,j=1}^{m}\mathbb{E}_{S,\sigma}\bar{a}_i\bar{a}_j\sigma_i\sigma_j\phi(x_i)^T\phi(x_j) \tag{41}$$

$$\leq\sum_{i}\mathbb{E}_{S,\sigma}\bar{a}_i^2\sigma_i^2\|\phi(x_i)\|^2 \tag{42}$$

$$+\sum_{i,j:i=\pi(j)\text{or}j=\pi(i)}\mathbb{E}_{S,\sigma}\bar{a}_i\bar{a}_j\sigma_i\sigma_j\phi(x_i)^T\phi(x_j) \tag{43}$$

$$\leq B^2\sum_{i}\mathbb{E}_{S,\sigma}\bar{a}_i^2\sigma_i^2 \qquad (\because \mathbb{E}\bar{a}_i^2 = \mathbb{E}\bar{a}_i = 1 - p_\alpha) \tag{44}$$

$$+ B^2\sum_{i,j:i=\pi(j)\text{or}j=\pi(i)}\mathbb{E}_{S,\sigma}\bar{a}_i\bar{a}_j\sigma_i\sigma_j \tag{45}$$

$$\leq B^2 m(1-p_\alpha) + B^2\sum_{i,j:i=\pi(j)\text{or}j=\pi(i)}\mathbb{E}_{S,\sigma}\bar{a}_i\bar{a}_j \qquad (\because \mathbb{E}\sigma_i \leq 1) \tag{46}$$

$$\leq B^2 m(1-p_\alpha) + B^2 2m(1-p_\alpha)^2 \tag{47}$$

. Thus:

$$\frac{1}{2}\mathbb{E}_{S,\sigma}\|\sum_{i=1}^{m}\bar{a}_i\sigma_i\phi(x_i)\| \tag{48}$$

$$\leq\frac{B}{2}\sqrt{m(1-p_\alpha) + 2m(1-p_\alpha)^2} \tag{49}$$

Through similar reasoning, the first term yields:

$$\frac{1}{2}\mathbb{E}_{S,\sigma}\|\sum_{i=1}^{m}a_i\sigma_i(\phi(x_i) - \phi(x_i')\| \tag{50}$$

$$\leq\frac{1}{2}\sqrt{\sum_{i=1}^{m}\mathbb{E}_{S,\sigma}a_i^2\sigma^2\|\phi(x_i) - \phi(x_i')\|^2} \tag{51}$$

$$\leq\frac{1}{2}\sqrt{\sum_{i=1}^{m}\mathbb{E}_{S,\sigma}a_i^2\alpha^2} \tag{52}$$

$$=\frac{1}{2}\sqrt{mp_\alpha\alpha^2} \qquad (\because \mathbb{E}a_i^2 = \mathbb{E}a_i = p_\alpha) \tag{53}$$

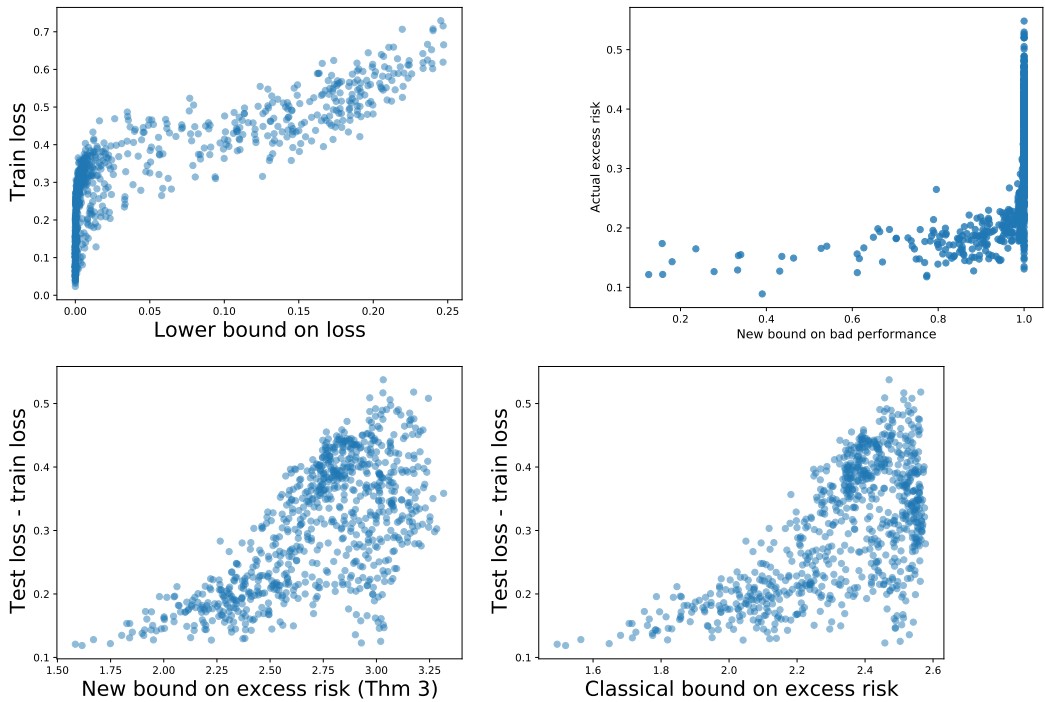

Figure 6: Scatter plots showing the correlations described in the main paper. For these scatter plots, data across all 45 tasks was pooled together. Clockwise from top left: Lower bound vs training loss, bound from theorem 2 vs excess risk, classical Rademacher-based bound vs excess risk, and Theorem 3 vs excess risk.

Putting these expressions back in equation 37, we get:

$$\mathcal{R}_m(\mathcal{H}_\phi) \leq \frac{W}{2m}(\sqrt{mp_\alpha\alpha^2} + B\sqrt{m(1-p_\alpha) + 2m(1-p_\alpha)^2} + B\sqrt{m(1-p_\alpha) + 2m(1-p_\alpha)^2})$$

(54)

$$= \frac{W\left(\alpha\sqrt{p_\alpha} + 2B\sqrt{m(1-p_\alpha) + 2m(1-p_\alpha)^2}\right)}{2\sqrt{m}}$$

(55)

$\square$

## A.2 Scatter plots

See Figure 6.

## A.3 Representations used in experiments

In general it is difficult to do statistically rigorous experiments for choosing representations since publicly available pre-trained representations vary wildly in architecture and dimensionality. However, we want to ensure that the characterization we produce is nuanced enough to distinguish between fairly similar representations that are based on the same backbone architecture and the same feature dimensionality. Therefore we used multiple feature representations that are all trained on ResNet 18 on the same 3 datasets (ImageNet, CIFAR-100 and iNaturalist) but with very different loss functions. These representations were trained in the context of an unrelated exploration on the use of coarsely labeled data (see supplementary). For completeness, we present a brief overview of these representations here.

All representations were trained on a combination of a coarsely labeled dataset $C$ and a finely labeled dataset $F$. $C$ may have some classes that are not present in $F$. The representations we used are:

1. A representation trained with simple supervised learning on $F$ alone.

2. A representation trained with simple supervised learning on $C$ alone.

3. A representation trained on $C \cup F$ with two classifier heads, one for coarse categorization and another for fine-grained categorization

4. A representation trained on $C \cup F$ with *fine-grained soft pseudo-labels* for $C$ obtained using a classifier trained on $F$

5. A representation trained on $C \cup F$ with *fine-grained soft pseudo-labels* for $C$ obtained using a classifier trained on $F$, and with the pseudo-labels filtered to be consistent with the coarse label,

6. A representation trained on $C \cup F$ with ground-truth fine-grained. labels for images in $C$.

6 representations trained on 3 datasets gives us 18 representations. Details on the training are in the paper linked in the supplementary. **These pre-trained representations will be released upon acceptance**.

### A.4 BOUNDS FOR MULTICLASS CLASSIFICATION

### A.4.1 PROBLEM SETUP

Suppose, as before, we are interested in mapping a space of inputs $\mathcal{X}$ to a space of targets $\mathcal{Y}$. There is an underlying distribution $\mathcal{D}$ over $\mathcal{X} \times \mathcal{Y}$. We have a feature representation $\phi : \mathcal{X} \to \mathbb{R}^f$ which might be pre-trained on another dataset or task, which is inaccessible to us. We will assume that $\|\phi(x)\| \leq B$.

For multiclass classification, our classifier will use a scoring function that scores how well a feature vector matches a class label, $h : \mathbb{R}^f \times \mathcal{Y} \to \mathbb{R}$. The predicted label will then be $\arg\max_y h(\phi(x), y)$, where ties are broken arbitrarily. The set of possible functions $h$ defines the hypothesis class for the classifier; denote this by $\mathcal{H}$. For most of our analysis, we primarily care about the *smoothness* of the functions in $\mathcal{H}$. We will assume that all functions in $\mathcal{H}$ are Lipschitz continuous in the first argument with Lipschitz constant less than $W$. Thus:

$$h(\phi(x), y) - h(\phi(x'), y) \leq W \|\phi(x) - \phi(x')\| \quad \forall h \in \mathcal{H}, \forall y \in \mathcal{Y} \tag{56}$$

We note that one such hypothesis class which is commonly used in practice is the class of *linear classifiers of bounded norm*: $\mathcal{W} = \left\{ \mathbf{v}, y \mapsto \mathbf{w}_y^T \mathbf{v}; \|\mathbf{w}_y\| \leq W \forall y \in \mathcal{Y} \right\}$.

For a data point $x$ with label $y$, the zero-one loss incurred by scoring function $h$ is $l^*(h, \phi(x), y) = \mathbb{I}[y \neq \arg\max_{y'} h(\phi(x), y')]$. We will use the following continuous margin-based upper bound:

$$l(h, \phi(x), y) = \min(1, \max(0, \max_{y' \neq y}(1 + h(\phi(x), y')) - h(\phi(x), y))) \geq l^*(h, \phi(x), y) \tag{57}$$

We can now prove equivalent lower and upper bounds as below.

### A.4.2 LOWER BOUND

**Claim 3.** *Consider two data points $(x, y)$ and $(x', y')$. If $\|\phi(x) - \phi(x')\| < \frac{1}{2W}$ and $y \neq y'$, then:*

$$l(h(\phi(x)), y) + l(h(\phi(x')), y') \geq 1 \tag{58}$$

*Proof.* Observe first that due to Lipschitz continuity of $h$:

$$|h(\phi(x), y) - h(\phi(x'), y)| \leq W \|\phi(x) - \phi(x')\| \leq \frac{1}{2} \quad \forall y \tag{59}$$

$$\Rightarrow h(\phi(x'), y) \in [h(\phi(x), y) - \frac{1}{2}, h(\phi(x), y) + \frac{1}{2}]] \tag{60}$$

This in turn implies that for any pair of classes $y_1$ and $y_2$,

$$h(\phi(x'), y_1) - h(\phi(x'), y_2) \geq h(\phi(x), y_1) - \frac{1}{2} - (h(\phi(x), y_2) + \frac{1}{2}) \geq h(\phi(x), y_1) - h(\phi(x), y_2) - 1 \tag{61}$$

Now, denote $l(h, \phi(x), y)$ by $a$ and $l(h, \phi(x'), y')$ by $b$. We will prove this claim by contradiction. Suppose, if possible, that the clam is not true, and $a + b < 1$. Since the loss is always non-negative, it follows that $a < 1$ and $b < 1$. Thus:

$$a = \max(0, \max_{y'' \neq y}(1 + h(\phi(x), y'')) - h(\phi(x), y)) \tag{62}$$

$$\geq \max_{y'' \neq y}(1 + h(\phi(x), y'')) - h(\phi(x), y) \tag{63}$$

$$\geq 1 + h(\phi(x), y') - h(\phi(x), y) \tag{64}$$

$$\Rightarrow h(\phi(x), y) - h(\phi(x), y') \geq 1 - a \tag{65}$$

Taking the other data point:

$$b = \max(0, \max_{y'' \neq y'}(1 + h(\phi(x'), y'')) - h(\phi(x'), y')) \tag{66}$$

$$\geq \max_{y'' \neq y'}(1 + h(\phi(x'), y'')) - h(\phi(x'), y') \tag{67}$$

$$\geq (1 + h(\phi(x'), y)) - h(\phi(x'), y') \tag{68}$$

$$\geq 1 + h(\phi(x), y) - h(\phi(x), y') - 1 \qquad (\because equation\ 61) \tag{69}$$

$$= h(\phi(x), y) - h(\phi(x), y') \qquad \geq 1 - a(\because equation\ 65) \tag{70}$$

$$\Rightarrow a + b \geq 1 \tag{71}$$

thus yielding a contradiction. $\square$

This claim directly leads to the following lower bound for the multiclass case, with the exact same proof:

> **Theorem 6.** *Let $l$ be the loss function defined above, and $\mathcal{H}$ be a hypothesis class of Lipschitz functions with Lipschitz constant at most $W$. Let $\mathcal{D}$ be a distribution over $\mathcal{X} \times \mathcal{Y}$, and $p_a^\phi$ and $p_c^\phi$ defined as above. Then*
>
> $$\inf_{h \in \mathcal{H}} \mathbb{E}_{x,y \sim \mathcal{D}}[l(\mathbf{h}(\phi(x)), y)] > \frac{1}{2}\left(1 - p_a^\phi\left(\frac{1}{2W}\right)\right) p_c^\phi\left(\frac{1}{2W}\right) \tag{72}$$

Note that the only difference between the bounds for the binary and the multiclass case is the factor $2W$ in lieu of $W$.

### A.4.3 UPPER BOUND

We can similarly give the following upper bound. We first begin with the following claim:

**Claim 4.** *For the multiclass loss function defined above, and if $h$ is $W$-Lipschitz as described above,*

$$l(h, \phi(x), y) - l(h, \phi(x'), y) \leq 2W\|\phi(x) - \phi(x')\| \tag{73}$$

*Proof.* We note that $l(h, \phi(x), y) = f(\max_{y' \neq y}(1 + h(\phi(x), y')) - h(\phi(x), y))$, where $f$ is defined in Theorem 4. Theorem 4 directly yields:

$$l(h, \phi(x), y) - l(h, \phi(x'), y) \tag{74}$$

$$\leq \left|\left(\max_{y' \neq y}(1 + h(\phi(x), y')) - h(\phi(x), y)\right) - \left(\max_{y' \neq y}(1 + h(\phi(x'), y')) - h(\phi(x'), y)\right)\right| \tag{75}$$

$$= \left|\left(\max_{y' \neq y}(1 + h(\phi(x), y')) - \max_{y' \neq y}(1 + h(\phi(x'), y'))\right) + (h(\phi(x'), y) - h(\phi(x), y))\right| \tag{76}$$

$$\leq \left|\max_{y' \neq y}(1 + h(\phi(x), y')) - \max_{y' \neq y}(1 + h(\phi(x'), y'))\right| + |h(\phi(x'), y) - h(\phi(x), y)| \tag{77}$$

$$\leq \max_{y'}|h(\phi(x), y') - h(\phi(x'), y')| + |h(\phi(x'), y) - h(\phi(x), y)| \tag{78}$$

$$\leq W\|\phi(x) - \phi(x')\| + W\|\phi(x) - \phi(x')\| \tag{79}$$

$$\leq 2W\|\phi(x) - \phi(x')\| \tag{80}$$

$\square$

The above claim then yields the following theorem

---

**Theorem 7.** *Suppose $S$ is a sampled training set of $m$ points. For any $h \in \mathcal{H}$, let $l_{max}(h, S) = \max_{(x,y) \in S} l(h, \phi(x), y)$ be the maximum loss $h$ incurs on $S$. Then, for all $\epsilon > 0$ and $(x, y) \sim \mathcal{D}$:*

$$P\left(l(h, \phi(x), y) > l_{max}(k, S) + \epsilon\right) \leq \left(1 - p_a^\phi\left(\frac{\epsilon}{2W}\right) p_c^\phi\left(\frac{\epsilon}{2W}\right)\right)^m. \tag{81}$$

---

*Proof.* As we have shown above, for all $x, x', y$:

$$l(h, \phi(x), y) - l(h, \phi(x'), y) \leq 2W \|\phi(x) - \phi(x')\| \tag{82}$$

It follows that for any $(x, y), (x', y')$:

$$\left.\begin{array}{c} \|\phi(x) - \phi(x')\| < \frac{\Delta}{2W} \\ y = y' \end{array}\right\} \Rightarrow l(h, \phi(x), y) \leq l(h, \phi(x'), y') + \Delta \qquad \forall h \in \mathcal{H} \tag{83}$$

Note that the probability of sampling such an $(x', y')$ is $p_a^\phi(\Delta/2W) p_c^\phi(\Delta/2W)$. If $(x', y')$ is in $S$, then $l(h, \phi(x'), y') \leq l_{max}(h, S)$. Thus, for any $(x, y) \sim \mathcal{D}$

$$P\left(l(h, \phi(x), y) \leq l_{max}(h, S) + \epsilon\right) \geq P\left(\exists (x', y') \in S \text{ s.t } \|\phi(x) - \phi(x')\| < \frac{\Delta}{2W} \text{ and } y = y'\right) \tag{84}$$

$$= 1 - \left(1 - p_c^\phi\left(\frac{\epsilon}{2W}\right) p_a^\phi\left(\frac{\epsilon}{2W}\right)\right)^m \tag{85}$$

$\square$

Once again, the only difference in proofs is the use of factor $2W$ instead of $W$.

