# OpenReview forum: "A theoretically grounded characterization of feature representations "
_ICLR.cc/2022/Conference — ICLR 2022 Submitted_

### Official Review · Reviewer_tEiy · 2021-10-26

**Correctness:** 4
**Technical Novelty And Significance:** 2
**Empirical Novelty And Significance:** 2
**Recommendation:** 6
**Confidence:** 3

**Main Review:**

This paper is clearly written, and the bounds are fairly intuitive and well presented. However, I think that the paper would be improved with better experimental evaluation, and more discussion of the existing literature.

Experimental evaluation
-------------------------

* The authors restrict their analyses to convenient representations they have from other work. While I share the author's desire for statistically rigorous experiments, the paper would be made much stronger if the authors demonstrated the validity of their approach with some existing representations, especially trained in different paradigms e.g. unsupervised. Even if the authors can't find multiple trained versions of the same architecture, just showing that their metrics e.g. predict how well SimCLR representations perform on different downstream tasks would improve the paper substantially. This would still allow a controlled comparison, since the source representations would be the same; it would just add another dimension of experiments to complement the present ones.

* Relatedly, this paper does not compare to the work of Nguyen et al. (2020) because it "assumes that representations come from classification tasks." This seems like a rather weak justification, since the present paper only evaluates representations pre-trained on classification (even if they sometimes do joint coarse + fine classification, one could apply the method of Nguyen et al. to the joint labels I believe). There may be other justifications for why the present method is superior (e.g. making use of unlabelled data); but demonstrating such benefits experimentally would improve the paper.

* While it can be useful to have transfer bounds that don't depend on training, computing the probabilities used in the bounds exactly involves pairwise comparisons, i.e. $O(N^2)$ computations over the data. For large supervised datasets it seems like the transfer estimates might be better from spending an equivalent amount of compute actually training a classifier on the data and seeing how well it did. I believe similar strategies are used in neural architecture search, because performance after e.g. one epoch of training is fairly strongly correlated with asymptotic performance. It would be useful for the paper to compare to such methods.

* I would also like to see some more analysis of how important the unlabelled data is in the few-shot setting, as this is fairly non-standard in the meta-learning literature. For example, the authors could vary the amount of unlabelled data, and plot the % accuracy drop correspondingly.


Literature
-----------

This paper would benefit from situating further within the existing theoretical literature on transfer. This work seems in several places to overlook prior theory work that considers generalization in ways that could be (or are) applied to understanding representations and transfer. Some of these papers may not conflict with this work or impact its novelty, but nevertheless might still be worth discussing. I've ordered the works that seem related to me in roughly what I see as the order of importance for addressing them.

* Petina & Lampert (2014) consider lifelong learning from a PAC-Bayesian perspective, including a representation transfer setting similar to that considered here. It would be useful to discuss any relationship between the bounds they express (taking the one-prior-task case from their work) and the present work.

* Arora et al. (2019, distinct from Arora's work cited in the paper) explore a "data-dependent complexity measure" that discriminates at least between random and true label distributions on real tasks, and is based on a characterization of failure probability in terms of Lipschitz functions. I haven't read this paper in detail, but superficially it seems very relevant to the present work, and it should probably be discussed.

* Cohen et al. (2020) consider object representation manifolds, and explore classification capacity in terms of the structure, extent, and relationships between these manifolds. These seems quite related to the notions used in this paper.

* Galanti et al. (2016) extend PAC bounds to the case of transfer and consider a few different cases. Their notion of transfer is a little different than explored in this work, since they consider simultaneous training with shared weights rather than classification of trained representations. However, it seems worth discussing.

* By making simplifying assumptions of linear hidden layers Advani et al. (2020) and Lampinen & Ganguli (2019) analyze learned representations and generalization in deep networks. The latter also consider some kind of transfer between tasks, though again shared weights rather than classification on trained representations. These results seem somewhat relevant, although they are within a simplified regression setting rather than classification.

* Schwartz-Ziv & Tishby (2017) explore representations in terms of their mutual information with the inputs and outputs, and the dynamics of how this MI evolves over training. While MI is less directly relevant to transfer, it could at least provide a lower bound th1at might be worth considering (i.e. transfer is impossible if no mutual information), and this work suggests it might be interesting to consider transferability partway through learning (e.g. perhaps representations are more transferable before the "compression" phase the authors describe).

It's also quite possible that there are even more relevant papers I've missed; I hope the other reviewers will share some.

References
-----------

Advani, M. S., Saxe, A. M., & Sompolinsky, H. (2020). High-dimensional dynamics of generalization error in neural networks. Neural Networks, 132, 428-446.

Arora, S., Du, S., Hu, W., Li, Z., & Wang, R. (2019, May). Fine-grained analysis of optimization and generalization for overparameterized two-layer neural networks. In International Conference on Machine Learning (pp. 322-332). PMLR.

Cohen, U., Chung, S., Lee, D. D., & Sompolinsky, H. (2020). Separability and geometry of object manifolds in deep neural networks. Nature communications, 11(1), 1-13.

Galanti, T., Wolf, L., & Hazan, T. (2016). A theoretical framework for deep transfer learning. Information and Inference: A Journal of the IMA, 5(2), 159-209.

Lampinen, A. K., & Ganguli, S. (2019). An analytic theory of generalization
dynamics and transfer learning in deep linear networks. International Conference
on Learning Representations

Pentina, A., & Lampert, C. (2014, June). A PAC-Bayesian bound for lifelong learning. In International Conference on Machine Learning (pp. 991-999). PMLR.


Shwartz-Ziv, R., & Tishby, N. (2017). Opening the black box of deep neural networks via information. arXiv preprint arXiv:1703.00810.



**Summary Of The Paper:**

This paper proposes to study how feature representations are transferable to downstream tasks. It presents a theoretical characterization of such transfer, in terms of relatively intuitive concepts of congregation and alignment. Specifically, it presents various bounds on classifier's expected error, probability of high-error inputs, and Rademacher complexity. The paper validates these theoretical observations by experiments with transfer in visual settings, between supervised tasks, or to few-shot (+unlabeled data) transfer.

**Summary Of The Review:**

The paper is clear and seems moderately useful; broader experimental evaluation and more engagement with the literature would improve it substantially.

---

> ### Author Response · Authors · 2021-11-22
> **Thank you**
>
> We thank the reviewer for taking the time out to provide us such detailed feedback.
> *Literature*: Thanks! We have included all these papers in our literature review. In general, we find our work to be similar to work on representing classes as manifolds, for instance, but with simpler and (we believe) more intuitive measurements of the feature space.
>
> *Self-supervised*: We now include some exemplar analysis of self-supervised representations in Table 1 and Figure 3. A more extensive evaluation will be done for camera-ready; we did not have time to perform this in the rebuttal period.
>
> *Large datasets*: When it comes to decision-making, we agree with the reviewer that our approach is computationally more expensive on large datasets. Practically, we believe our approach is more useful for few-shot problems where training classifiers is not possible or might yield very noisy results. Hence our focus on few-shot in the experiments. That said, even on large datasets, our bounds are more intuitive than simply looking at classifier accuracies, which may not reveal a satisfying explanation for why one representation is better than the other.
>
> *Unlabeled data*: This is indeed an interesting experiment. Unfortunately, we ran out of computational resources to perform this analysis for the rebuttal; we will provide this for the camera-ready.

---

> > ### Comment · Reviewer_tEiy · 2021-11-24
> > **Thanks for the response, I do feel the paper has improved somewhat.**
> >
> > Thanks to the authors for their response, I do feel that the paper has improved.  However, I do not think the improvement merits a full movement from a 6 to 8 rating. It is unfortunate that we cannot give 7 ratings this cycle, but as it is I am not revising my score. I do recognize that the time to respond to reviews was quite short, and I feel like the experiments the authors propose to perform before the camera-ready deadline would improve the paper further if they were well-integrated into the discussion. I also continue to believe that the prior literature could be better discussed; while I appreciate the authors making the effort to reference the literature I noted, the statements they make seem relatively uninformative to the reader beyond noting that this literature exists. For example "Our proposed measurements are similar, but are simpler to measure and potentially more intuitive."—the reader might benefit much more from a few more sentences about in what ways the measurements are similar and different, *why* they are simpler to measure and more intuitive, and if there are any aspects of the prior work that might be positive in some settings.

---

> > > ### Author Response · Authors · 2021-11-24
> > > **Thank you again!**
> > >
> > > We will improve the related work discussion further in the camera ready.

---

### Official Review · Reviewer_PXiU · 2021-11-01

**Correctness:** 3
**Technical Novelty And Significance:** 3
**Empirical Novelty And Significance:** 3
**Recommendation:** 6
**Confidence:** 3

**Main Review:**

Defn 1: This should be defined more carefully, with (x,y)(x',y') ~ D. Also, not
e that this quantity could be defined conditional on x.

Defn 2: This should be defined more carefully, with x, x' ~ D.  Figure 1 contrasts lower (left 2 plots) and higher (right 2 plots) degree of concentration. However, as defined, p_c^{\phi}(\alpha) would depend on the scaling of the feature space. Although at the start of sec 3 we are told ||phi(x)|| \le B, this is never used below, so in fact this definition just serves to set the scale of the
 feature space. For example, it is easy to see that for a Gaussian p(x) in 1-d,
 this quantity will depend on the standard deviation of the distribution (and it is easy to extend the analysis to a multivariate Gaussian.) I suspect that the notion that is wanted here the entropy of p(x) relative to a Gaussian with the same mean and covariance, i.e. the negentropy https://en.wikipedia.org/wiki/N
egentropy , as the Gaussian has maximum entropy for a given mean and covariance
.

Notice also that a Lipshitz constraint only makes sense relative to the scale of the feature space. If we double its scale, then we need to half the Lipshitz
constant to retain the same semantics.

Comment: I believe that the intuition behind Defn 1 is relevant to the issue of
 "what makes a feature representation good for a target task?", but am rather unconvinced by the subsequent analyses, for reasons explained below.


Claim 1: The derivation of eq 7 from eq 6 is incorrect -- the inequality in eq
6 could also imply h(phi(x)) \le 1 + h(phi(x')) .  However I believe the claim in eq 5 is correct. Consider the functions \ell(h,+1) and \ell(h,-1) as a function of h. Eq 6 says that |h(phi(x) - h(phi(x')| \le 1, so we can "slide" one of these functions in h by (-1,1) relative to the other. The result is then intuitive.

However, I doubt that Claim 1 is of great practical use. The loss function (eq 2) is introduced because "the zero-one loss ... is difficult to analyze". However, Claim 1 depends critically on the "margin" for l(h,\pm 1) extending to h=\pm 1, and indeed the Lipshitz condition is specifically designed to exploit this margin to obtain eq 5. Thus I don't believe that this property is relevant to the zero-one loss we actually care about.

Theorem 1:

Written out more carefully, the required statement is Pr( ||phi(x) - phi(x')|| < 1/W AND y \neq y') =
Pr( ||phi(x) - phi(x')|| < 1/W) \times Pr( y \neq y') | ||phi(x) - phi(x')|| < 1/W) .

We start my making the above statements *conditional on x* as mentioned for Defns 1 and 2 above, giving Pr( ||phi(x) - phi(x')|| < 1/W AND y \neq y' | x) = p^{phi}_c(alpha;x) \times (1 - p^{phi}_a(alpha;x)) , where the notation ;x indicates dependence on a particular x.

We can now try to average the expression p^{phi}_c(alpha;x) \times (1 - p^{phi}_a(alpha;x)) wrt D ~ x but an issue arises with the product term E_x [p^{phi}_c(alpha;x) p^{phi}_a(alpha;x)] which cannot be reduced to the product of the individual expectations.

Hence I believe there is a "bug" in this derivation for Theorem 1 as stated.

Another issue is that Theorem 1 depends only on the losses obtained when ||phi(x) - phi(x')|| < 1/W AND y \neq y', so other losses will be sustained for situations where this condition does not apply, so the bound may be very loose. Also as noted above it depends critically on the loss function used in eq 2, and so may have little relevance to the zero-one loss of interest.

Experiment 7.1: I am unclear if the loss you are reporting on the y-axis is the loss as per eq 2, or 0/1 loss -- please clarify. Also you don't state the alpha used (or how it was chosen), or the size of the training and test sets -- please expand (remember -- reproducible science!).

I realize that this is mainly a theory paper, but I believe the analysis of the experiments can be much improved. For a start you are only carrying out experiments on one dataset -- while for a more experimental paper we would probably look for say 10 datasets.

For the CIFAR-10 data, for each of the 45 tasks we have 18 different representations. For EACH of the 45x10 trials, one can compare the training and test losses for each of the 18 representations against the bounds. Strong evidence for the efficacy of the bound would be that the ordering of the observed losses would correlate well with the ordering of the bounds, over most of the 45x10 replications. So for example you could compute your Spearman's rho on the 18 datapoints, and make a histogram of the values obtained over the 450 expts.  As it is, such a fine-grained comparison is "washed out" in the plots in Fig 3.

The claim that a rho of 0.61 is much better than 0.56 is, I'm sorry to say, pretty silly -- just look how similar the plots in Fig 5 are.

Sec 7.2 considers a few-shot task -- here for model selection the new bounds are shown to give better performance than the H-score and TransRate. It is notable that these last two methods can perform worse than the Random baseline.

Further comments: It is worth noting that the "purity" of the classes in a given ball of radius alpha around phi(x) (as in Defn 1) is something that also arises in the "RadiusNeighborsClassifier" version of k-nearest neighbors, see e.g.  https://scikit-learn.org/stable/modules/neighbors.html#nearest-neighbors-classification -- for this paper the radius is given by alpha.


Summary: While the intuition behind Definition 1 makes quite a lot of sense, there seem to be some errors in the derivation of Theorem 1. Also, importantly, it is not so clear that bounds on the loss from eq 2 will be well correlated with the 0/1 loss. (This could be investigated experimentally in sec 7.) The experiments could be carried out more thoroughly (as described above) to gain a better understanding of what is going on. Overall, my judgment is that this paper is not of ICLR standard.

** Other points

Eq 18: Is the 1/2W in eq (18) a typo for 1/W?

Eq 24: enhance readability by using larger brackets the P() statement on the LHS, as per the RHS.

p 7 para 2, defn of Rademacher complexity should contain \sum_i \sigma_i h(x_i) not \sum_i h(x_i)

== post rebuttal ===

I have already made comments on revised manuscript  (on 27 Nov) and the authors have responded to these.

Pros:

* the idea of relating performance/generalization to p_a and p_c seems useful.

* I like the new Fig 3 (as suggested by some of the reviewers and now implemented.)

Cons:

* As I've made clear before, I believe Thm 1 is unlikely to be of much import, as it makes use of the specifics of the "margin" of the
  loss in eq 2. I'd like to see more focus on 0/1 loss in the expts,  rather than the margin loss. (I note 0/1 loss results are promised
  for the camera ready version.)

* As I understand it, Thms 2 and 3 are not particularly novel (my understanding here is from other reviewers, I am not a learning theory
expert.)

* As pointed out by many reviewers, there does not seem to be a good  justification for the use of 100*lower bound + upper bound as a
  basis for model selection.

I am sticking with my updated score of 6.





**Summary Of The Paper:**

The paper presents two properties, "local alignment" (eq 3) and "degree of congregation" (eq 4) which are claimed to be good predictors of downstream (classification) task performance. These properties are used to derive bounds on the error of downstream classifiers (under a number of assumptions, including a linea
r classifier with Lipshitz constraints), and are investigated empirically for pairwise classification tasks derived from the CIFAR-10 dataset, and for model selection on a few-shot task.

**Summary Of The Review:**

While the intuition behind Definition 1 makes quite a lot of sense, there seem to be some errors in the derivation of Theorem 1. Also, importantly, it is not so clear that bounds on the loss from eq 2 will be well correlated with the 0/1 loss. (This could be investigated experimentally in sec 7.) The experiments could be carried out more thoroughly (as described above) to gain a better understanding of what is going on. Overall, my judgment is that this paper is not of ICLR standard.

==updated===
see post-rebuttal summary.

---

> ### Author Response · Authors · 2021-11-22
> **Thank you**
>
> We thank the reviewer for their exceptionally detailed comments.
>
> *Definitions*: Corrected. That said we keep the unconditional (or marginal) definitions for the reason described below.
>
> *Scaling of the feature space*:
> Yes, pa and pc depend on the scale of the feature space. But in current practice feature representations are normalized to have max norm 1 and the theorems and results will still hold (in fact our empirical analysis does this). The figures are explicitly chosen to show that even with fixed -norm features, these quantities are meaningful.
>
> *The derivation of eq 7 from eq 6 is incorrect* : The reviewer is *incorrect*: Actually it is not either-or, but both. That is, $|a -b| \leq 1$ means that $a> b-1$ *and* $a< b+1$. We only need the latter for the proof. Thus our claim is actually correct.
>
> *Practical use of claim 1*: It is relevant in practice for few-shot problems, where training examples can be far away from the decision boundary and as such get a zero margin loss. In that case, the test examples that are alpha away and with a different label are constrained to have a margin loss >=1, which actually implies that they are incorrectly classified.
>
> *Conditional vs unconditional definitions*: The purported “bug” in the theorem arises precisely because the conditional definition is used. We used the unconditional definition (or more precisely, the marginal distribution) exactly to avoid this bug. We agree with the reviewer that a conditional treatment would lead to better results; however, it is much more difficult to succinctly characterize and measure in practice.
>
> *Loss, alpha in plots*: Corrected.
>
> *Loose lower bound*: Yes, the lower bound can be quite loose, which is why we also provide an upper bound. Our few-shot experiments indicate that in fact in conjunction, the two losses do provide a significant enough correlation to allow us to choose feature representations.
>
> *Correlations*: Thank you for your suggestion! We have replaced the scatter plots with the fine-grained analysis you describe. We still averaged over 10 trials because each individual trial can be exceptionally noisy. However, for each of the resulting 45 tasks, we now compute separate correlation values and plot a histogram.
>
> *Radius neighbor classifier*: Yes, we agree, and future work might consider such classifiers instead.

---

### Official Review · Reviewer_J9Rs · 2021-11-02

**Correctness:** 3
**Technical Novelty And Significance:** 4
**Empirical Novelty And Significance:** 4
**Recommendation:** 8
**Confidence:** 2

**Main Review:**

strengths:

- The paper is very well written. The exposition is clear. The introduction makes very clear the specific challenges to be addressed in the paper and the approach taken.
- The description of the relationship to previous work is thorough and precise.
- Addresses an important question

weaknesses:

- limitations not well acknowledged/discussed. (As in any work of this sort) the presented results address a question that is narrower in scope than the one posed in the introduction. For example, the analysis assumes the relevant task is a classification task and so the results may not be relevant for reconstruction tasks or other regression tasks. The assumption that the scoring function is W -Lipschitz narrows what is meant by "good representation". It seems to imply that a good rep is one that is already appropriately disentangled, rather than one that can be disentangled by, say, a neural network (which would violate the assumption), to do something useful (or several useful things). A small point.
- Pearson seems more appropriate than Spearman correlation here since the the losses and errors are on an interval rather than ordinal scale.
- No statistical tests or confidence intervals on the correlations presented in the results.

minor comments:

- missing primes on x and y in second loss in the text between equations 7 and 8
- would be good to know to what extent your correlations are driven by the concentrations at the extremes in Figure 3.
- I don't think you need the line "Theoretical bounds are useless if they do not impact practical decisions."
- Not clear what representations were tested in the experiments without consulting the appendix. Consider moving a basic description to the main text.
- Consider adding subpanel titles (CUB-200, CIFAR-10) to Figure 4 to make it easier to grok. You could also include a more informative plot that a bar plot, e.g. show the distribution with box & whisker or violin plot, etc.

**Summary Of The Paper:**

The paper addresses the question of what makes a representation suitable or "good" for a particular task. The submission includes three data-dependent bounds: a lower bound on the average accuracy and two upper bounds on generalization. The analysis is based on the simple and intuitive concepts of local alignment and degree of congregation of the data points within a representational space and according to a binary labeling of the points. Two experiments support the theoretical claims.

**Summary Of The Review:**

The submission seems like a clear accept. It makes a clear contribution to an important topic and is very well written and clearly described. The statistical analysis of the empirical results could be improved. I made a number of minor comments that I think can be easily addressed to improve the quality of the paper. I have no major complaints.

---

> ### Author Response · Authors · 2021-11-22
> **Thank you**
>
> We thank the reviewer for the thoughtful comments.
> *Limitations*: The reviewer is exactly right about the limitations. We have replaced the conclusion with a discussion of these limitations. We do note that the assumption of a linear classifier is central to much of representation learning work, such as self-supervised learning.
>
> *Pearson vs Spearman*: While the reviewer is right that the values are not ordinal, we chose Spearman because we explicitly wanted to see if our bounds rank representations in an informative manner. Thus it serves as a motivation for our few-shot transfer results, which shows that you can use the bounds to choose representations. We fully expect that the actual bounds might be loose; we are primarily concerned (in the correlation computations) with whether they are predictive or not.
>
> *Correlations*: Following the suggestions of reviewer PXiU, we have replaced scatter plots with a histogram of the correlations obtained in each of the 45 different tasks. We believe that this gives a more fine-grained picture of the correlation. (Unfortunately, with the pooled data, we had enough data points for all correlations to be statistically significant).
>
> *Figure 4*: Thanks for the suggestion! we have now converted this to a violin plot with panel titles to show the results more clearly.

---

> > ### Comment · Reviewer_J9Rs · 2021-11-28
> > **Issues remain, but are addressable**
> >
> > Thank you for the response. I appreciate the addition of the discussion of limitations. The response to the Pearson vs Spearman question seems reasonable.
> >
> > I don't fully understand the response to the point about hypothesis testing. As far as I can tell from the updated text, no hypothesis tests have been performed to quantify the superiority of the proposed bounds. I agree that the histograms in Figure 4 show the result well. But still, the effects/differences that you describe in Fig 4 and 5 are not quantified. For example, is your error rate significantly better than TransRate on CUB 200? I don't insist on p values but I think stated differences/effects should be quantified with an effect size or confidence interval. Reviewer PXiU has also made this point.
> >
> > I agree with reviewer sCcc that the ideas formalized here are intuitive, but I don't think that is cause for rejection unless we can point to previous work formalizing those intuitions. Rather, that the ideas are intuitive should be considered a strength.
> >
> > Reviewer PXiU's question about reporting the 0/1 loss rather than the margin loss seems reasonable.
> >
> > Several other reviewers identified issues that I missed on first reading. The paper has clearly been improved by the review process but it appears that more improvements could be made. However, most if not all remaining issues seem like they could be addressed for the camera-ready version. They are not fundamental. I would consider lowering my score to 6 to be in alignment with the other reviewers given the outstanding issues, but I'm still in favor of acceptance. I would keep my score as it is if the authors commit to making the proposed changes for the camera-ready version.

---

> > > ### Author Response · Authors · 2021-11-29
> > > **Response**
> > >
> > > Thanks for your response and your support!
> > >
> > > Re: p-values, using the Wilcoxon non-parametric test for paired samples and a p-value threshold of 0.05, we find that the accuracy obtained by models chosen by our approach is statistically significantly different from those based on *H-score * and Random on CUB. On CIFAR, it is statistically significantly different from H-score and TransRate. We will include these in the camera-ready.
> > >
> > > For Figure 4, we observed that the correlation between the lower bound and training error was statistically significant (p<0.01) for all of the tasks. The correlation between Theorem 2 and excess risk was statistically significant for ~95% of the trials. The correlation between Theorem 3 and excess risk was significant for ~73% of the trials, compared to 33% for the classical Rademacher based bounds. We will add these to the camera ready as well.

---

### Official Review · Reviewer_sCcc · 2021-11-03

**Correctness:** 4
**Technical Novelty And Significance:** 2
**Empirical Novelty And Significance:** 3
**Recommendation:** 5
**Confidence:** 3

**Main Review:**

Strength: the paper is clearly written and easy to follow. The observations make intuitive sense.

Questions:
  - How does the correlation between your lower bound and test loss compare with the correlation between kNN accuracy and the test loss?
  - How are the values of $p_{a}^{\phi}, p_{c}^{\phi}$ distributed empirically?
  - Empirically, how well can the lower bound predict the performance, when the learned representations are far from being clustered (for example, on representations from some lower layer of a network)?
  - For few-shot transfer, how would the performance change if we vary the number of training samples per class?
  - Do you have any recommendations on how to increase $p_{a}^{\phi}$ and decrease $p_c^{\phi}$ in practice?

Minor point: in equation (24), $\Delta$ should be $\epsilon$.



**Summary Of The Paper:**

This work studies the relation between properties of the feature extractor, and the downstream performance on binary classification tasks.
The authors identify two key properties, namely "local alignment" and "congregation", which can be used to derive lower bound on the test accuracy and generalization bounds.
The theoretical results are verified empirically on CIFAR10 and CUB200.

**Summary Of The Review:**

The paper is clear and well justified.
However, I find the results unsurprising, since local alignment and congregation together mean that the features should be well clustered, which is intuitive and well known.
Personally I think the paper would need more technical contributions to be accepted.

=== Post rebuttal update ===

Based on the authors clarifications and additional experimental results, I have raised my score to 5. While I agree that the paper has improved after the revision and I acknowledge the contributions, it's still below the bar to me though only marginally.

Specifically, I am now convinced that the intuitive connection with knn is a plus, though I still think the theoretical novelty is limited.
Given the theoretical contribution, I'd expect stronger empirical results for the proposed method to be convincing. Potential improvements, as also mentioned by other reviewers, include 1) comparing with more baseline methods with a larger variety of features (e.g. expand Table 1 with results from other self-supervised methods, including non-contrastive ones), and 2) better justifying the criteria for model selection (i.e. the sum of the lower bound in Thm 1 and the upper bound in Thm 2) as it currently seems arbitrary.

---

> ### Author Response · Authors · 2021-11-22
> **Thank you**
>
> We thank the reviewer for their detailed review and their constructive comments.
>
> *Unsurprising results*: Our results are actually two fold:
> - if we have infinite training data, actually we don't need features to be congregated at all. So we don't need clustering.
> - we need clustering only for generalization from small training sets.
> The first conclusion is in fact counter-intuitive and explains the success of contrastive self-supervision that does not attempt to do clustering at all. Per the other reviewers’ comments, we now have shown this effect of self-supervision in Table 1.
>
> *Correlation between knn performance and test loss*: While we did not compute this correlation due to computational constraints, the reviewer is right in their view that the knn performance will correlate with the test loss. This should be clear from the fact that (a) knn is Bayes optimal, and (b) our analysis is very similar to what is done by a knn classifier. However, computing the knn accuracy requires a large labeled training set, and so the knn accuracy cannot be used for few-shot model selection, as in our experiments.
>
> *Empirical distribution of pc, pa*: Figure 3 shows how pc and pa are distributed empirically for three feature representations: a random network, an ImageNet-trained network and a network trained with self-supervision. As expected, pa falls and pc rises with alpha, but the rate and fall depends on the representation.
>
> *Predicting performance when representations are far from clustered*: In fact, many pre-trained networks are not clustered at all, and have low values of congregation (see Figure 3 in the updated paper). The lower bound then predicts low error, which actually predicts very well the accuracy of a classifier trained with a large training sets. For accuracy obtained with smaller training sets, the generalization bound in Theorem 2 yields more useful predictions. For these reasons, our experiments on few-shot transfer actually used a combination of the two bounds.
>
> *Few shot transfer with more data* : With more training samples, estimates of p_a get better, but differences between feature representations are much smaller.
>
> *Recommendations for increasing pa and decreasing pc*: It is not clear you want to decrease p_c, since that will hurt generalization from limited data (theorem 2). But if one does want to decrease p_c, contrastive learning does this (as shown in Table 1). Contrastive learning techniques also push together feature representations of augmentations of the same image, which may have the effect of increasing p_a. Unfortunately, defining the right training objective that uses these results is difficult, because it also depends on the training dynamics.

---

> > ### Comment · Reviewer_sCcc · 2021-11-26
> > **Thank you for your response**
> >
> > Thank you for the clarifications, especially on kNN and clusterability; I agree with the authors that being linearly classifiable doesn't require the features to be clustered.
> >
> > I appreciate the added empirical results in the revised draft. My initial impression was that this is mainly a theory paper, however I didn't find the results or proof techniques to be surprising or novel. I'd suggest to remove proofs from the main text and focus on empirical implications instead.
> >
> > I have raised the score on empirical novelty and the overall score. I'm willing to be convinced to raise the score further, but I'd like to have some clarifications on the new results:
> >   - Is the recommendation that we should use the lower bound for model selection? The upper left subplot in Figure 4 only shows the correlations with the _training_ loss, but not the test loss which is what we care about; for example, the authors mentioned that the Spearman's $\rho$ is 0.55 which is not very strong. I'd like to have the authors' comment regarding this.
> >   - Figure 5: the random baseline seems pretty competitive, could the authors comment on this please?
> >   - Figure 6, upper right subplot: are data points here the same as the ones used for the correlation histogram in Figure 4 (upper right subplot)? The upper bound (Thm 2; x-axis) in Figure 6 seems to be loose, even giving a trivial bound of 1 many of the times, and I'd like to have the authors' comment on this.
> >
> > As a side note, perhaps consider moving empirical results on the Rademacher upper bound to the appendix if short on space: the new bound in Thm 3 and the classical bound (i.e. bottom rows of Fig 4 and Fig 6) do not have a clear comparison, and both seem pretty poorly correlated with the excess risk, which is expected since Rademacher bounds are known to be loose.

---

> > > ### Author Response · Authors · 2021-11-29
> > > **Response**
> > >
> > > Thanks for your response!
> > >
> > > 1. *Lower bound for model selection?*: No, because the lower bound, as the reviewer correctly observes, does not consider generalization. For model selection, we used lower bound*100 + upper bound (although the selection does not seem to be that sensitive to this weight).
> > > 2. *Random baseline*: Indeed, the random baseline is competitive. However, this is more a reflection of the efficacy of model selection techniques, which is very poor. This is because model selection techniques assume the availability of labeled data, but we are in a *few-shot* setting where labeled data is scarce. This is especially problematic if one tries to measure class-conditional covariance matrices in high dimensional feature space. We will include this discussion in the camera-ready.
> > > 3. *Loose upper bound*: We apologize; we accidentally included an incorrect version of Figure 6 top right (it uses an incorrect value of W, the Lipschitz constant of the classifier). The data used for computing correlations is however correct. The correct plot was in the original paper version, Figure 3. This will be corrected in the camera ready.
> > >  That said, yes the upper bound can be loose. However, as our few-shot model selection results show, they are still useful for model selection.
> > >
> > > We thank the reviewer for the edit suggestions, and we will indeed do this to make room for some of these discussions in the final version.

---

> > > > ### Comment · Reviewer_sCcc · 2021-11-29
> > > > **Further clarification**
> > > >
> > > > Thank you for your response!
> > > >
> > > > For model selection, may I ask what's the reason behind using a weighted sum of the lower and upper bound? For example, is it because this combination works the best empirically (with the intuition that the upper bound in Thm 2 seems to correlate well with the risks), or is there a formal justification?

---

> > > > > ### Author Response · Authors · 2021-11-29
> > > > > **Clarification**
> > > > >
> > > > > The lower bound does not consider generalization at all (it bounds the error of the best possible classifier). The upper bound bounds the generalization error, but not the training error. Thus the two are capturing complementary sources of error suggesting that we must combine them.  The precise weight though is arbitrary.
> > > > >
> > > > > This linear combination is under "Experimental setup" in section 8.2,

---

### Official Review · Reviewer_DEHm · 2021-11-03

**Correctness:** 3
**Technical Novelty And Significance:** 3
**Empirical Novelty And Significance:** 2
**Recommendation:** 6
**Confidence:** 3

**Main Review:**

Minor Comments

The paper is well written. The concepts presented are intuitive and seem useful. And their transferability to the empirical tasks chosen seems well demonstrated. As an empiricist, I defer to other reviewers when it comes to reviewing the theoretical proofs, but I can speak to the paper’s claims of practical application.

My main concern is that the empirical tasks chosen are very constrained classification problems (made into binary tasks by choosing pairs of classes; no multi-class fine-grained classification tasks as is more realistic). I understand the desire to start with a simple settings, especially in a theoretical paper, but it makes it hard to justify the paper’s claims of testing “what makes a [good] feature representation” and “impact[ing] practical decisions”.

Another question: how does one know if an embedding is not “congregated” unless there is a less-congregated frame of reference? The paper alludes that “congregation” might help explain the success of contrastive learning, so it’s disappointing to see no experiments trying to demonstrate this.

I also believe the paper would have been improved if the authors explicitly encouraged/discouraged local alignment/congregation when optimizing models, and showing that this changes the metrics introduced accordingly.


**Summary Of The Paper:**

The paper asks the question “what makes a feature representation good for a target task?” In order to tackle this question, it introduces the concepts of local alignment (examples similar in feature space -> same labels), and congregation (how much examples generally embed close to each other). It uses them to produce bounds on achievable accuracy on binary classification problems which seem to be correlated with actual accuracy on real binary tasks.


**Summary Of The Review:**

This is a well-written paper that investigates a narrow task setting which unfortunately does not justify its claims of practical applicability. This could change with more extensive experiments in realistic datasets/settings.

---

> ### Author Response · Authors · 2021-11-22
> **Thank you**
>
> Thank you for your detailed review and constructive comments.
>
> *Multi-class classification*: The appendix now includes corresponding bounds for multiclass classification as well (section A.4). While we haven’t yet updated the experiments to include multiclass classification due to time constraints, we will do so for the camera-ready.
> Congregation with a frame of reference: In our definitions, this frame of reference is provided by the parameter $\alpha$. In the bounds, the Lipschitz constant of the classifier essentially provides us with a frame of reference, since such a classifier simply cannot resolve data points that have different labels but are embedded closer than a certain radius.
>
> *Self-supervised learning*: We now have experiments (see Table 1) showing that self-supervised methods do in fact substantially reduce the congregation in the underlying feature representation compared to a randomly initialized network. We also show that this results in much higher accuracy (38% reduction in loss) when lots of training data is available (as predicted by Theorem 1) but fails  to reduce the generalization error (<8% reduction) (as predicted by Theorem 2). Note that this result is even more starkly demonstrated by the ImageNet initialization, which is even less congregated.
>
> *Explicit control of pa and pc*: This is an interesting avenue for future work, but unfortunately not something we could rigorously test in the rebuttal period. That said, we assert that a measurable and predictive characterization of existing feature representations is useful in and of itself; the question of a training objective that uses this characterization requires further considerations of, e.g., training dynamics.

---

> > ### Comment · Reviewer_DEHm · 2021-11-29
> > **Response to authors**
> >
> > Thank you to the authors for their response. I think the addition of multi-class results improves the paper though it would be further improved with empirical results. I maintain my score of 6

---

### Author Response · Authors · 2021-11-22
**Summary of changes**

We thank the reviewers for exceptionally detailed reviews, and the area chair for recruiting an exceptional pool of reviewers. We appreciate the many suggestions for improvement. We have made the following changes:
- Added proofs and theorems for the multiclass case to the appendix (section A.4, per Reviewer DEHm)
- Added exemplar figures and tables showing how our bounds help understand self-supervised learning and its performance (Figure 3, Table 1, per Reviewer DEHm and Reviewer tEiy).
- Added a plot of empirically observed values of pa and pc (Figure 3, per Reviewer sCcc)
- Modified our correlation plot to provide fine-grained information by producing a histogram of correlations over tasks (Figure 4, per reviewer PXiU)
- Modified our few-shot results to show violin plots of relative errors (Figure 5, per reviewer J9Rs).
- Added a discussion of limitations (per Reviewer J9Rs).
- Added a more extensive literature review (per Reviewer tEiy).

Due to computational reasons, we were not able to do the following in time for rebuttal, but will do so in the camera ready:
- Compute correlations with k-nn accuracy (per Reviewer sCcc)
- Perform extensive experiments with self-supervised learning (per Reviewer DEHm and Reviewer tEiy).
- Include experiments with multiclass problems (per Reviewer DEHm)
- Repeat experiments in Section 8.2 with varying amounts of unlabeled data (per Reviewer tEiy)

---

### Decision · Program_Chairs · 2022-01-20

**Decision:**

Reject

**Comment:**

In this paper, authors introduce two properties of feature representations, namely local alignment and local congregation, and show how these properties can be predictive of downstream performance. The paper has a heavier focus on providing theoretical statements using these properties but authors also empirically evaluate their suggested method.

**Strong Points**:
- The paper is well-written and easy to follow.


- The proposed concepts (local alignment and local congregation) are intuitive.


- The theoretical statements and their proofs are correct.


- The proposed metric shows some advantage against a few baselines.


- Prior work on feature representations and transferability are discussed.


**Weak Points**:


- **The connections to prior work on K-nearest neighbors and linear classifiers are not properly discussed.** This is very important because authors assume that the network that outputs the feature representations is trained on a different data and they reduced the analysis to that of a binary linear classifier. Hence, all classical learning theory results on binary classifiers apply in this setting. Furthermore, KNN methods and analysis can be simply applied on the features as well. In light of this and the lack of discussion on this matter, the significance of the theoretical and empirical results are not clear.


- **The main proposed properties could be improved further**. It looks like the defined properties (local alignment and local congregation) could be improved by merging them into one property about separability of data? The current properties are sensitive to scaling which is undesirable given that classification performance is invariant to scaling of the features. It seems like local congregation is mostly capturing the scale so some normalized version of local alignment might be able to capture the main property of interest.


- **The theoretical results in their current form are not very significant.** One limiting factor on the theoretical results is that since the analysis is done only on the classification layer, it does not say anything about the relationship of the upstream and downstream tasks. But perhaps the most important limitation is that the properties are defined based on the downstream task distribution as opposed to downstream training data. That makes it difficult to measure them in practical settings where we have a limited number of data points. Classical results on learning theory avoid this and only use measures that depend on the given training set.


- **The empirical evaluation could benefit from stronger baselines** Authors mentioned "We therefore consider only baselines that make minimal assumptions about the pre-trained feature representation and the target task" and hence avoided comparing to many prior methods. However, I think the appropriate approach would be to compare the performance of the proposed method to strong baselines but then explain how they differ in terms of their assumptions, etc. Moreover, there are other simple heuristic baselines to consider, eg. K-NN (which is not computationally expensive in the few-shot settings) or a classifier that is trained by initializing it to be the sum of feature vectors in the first class (assuming binary classification) minus sum of feature vectors in the second class and doing a few SGD updates on it. Therefore, I believe authors could improve the empirical section significantly by taking these suggestions into account.


**Final Decision Rationale**:

This is a borderline paper. While the paper has a nice combination of theoretical and empirical contributions, both theoretical and empirical contributions have a lot of room for improvement (and a clear path to get there) as pointed above. In particular, I believe having either strong theoretical contributions or strong empirical contributions would have been enough for acceptance and I hope authors would take the above suggestions into account and submit the improved version of this work again!